# Longitudinal Characterization of a Neutralizing and Total Antibody Response in Patients with Severe COVID-19 and Fatal Outcomes

**DOI:** 10.3390/vaccines10122063

**Published:** 2022-12-01

**Authors:** Ricardo Serna-Muñoz, Alejandra Hernández-Terán, Maribel Soto-Nava, Daniela Tapia-Trejo, Santiago Ávila-Ríos, Fidencio Mejía-Nepomuceno, Emma García, Manuel Castillejos-López, Anjarath Lorena Higuera-Iglesias, Arnoldo Aquino-Gálvez, Ireri Thirion-Romero, Rogelio Pérez-Padilla, José Leopoldo Aguilar-Faisal, Joel Armando Vázquez-Pérez

**Affiliations:** 1Departamento de Investigación en Tabaquismo y EPOC, Instituto Nacional de Enfermedades Respiratorias Ismael Cosío Villegas INER, Mexico City 14080, Mexico; 2Escuela Superior de Medicina, Instituto Politécnico Nacional, Mexico City 11340, Mexico; 3CIENI Centre for Research in Infectious Diseases, National Institute of Respiratory Diseases, Mexico City 14080, Mexico; 4Departamento de Unidad de Epidemiología Hospitalaria e Infectología, Instituto Nacional de Enfermedades Respiratorias Ismael Cosío Villegas INER, Mexico City 14080, Mexico; 5Laboratorio de Biología Molecular, Departamento de Fibrosis Pulmonar, Instituto Nacional de Enfermedades Respiratorias Ismael Cosío Villegas INER, Mexico City 14080, Mexico

**Keywords:** SARS-CoV-2, neutralizing antibodies, severity, variants

## Abstract

The host immune response to SARS-CoV-2 appears to play a critical role in disease pathogenesis and clinical manifestations in severe COVID-19 cases. Until now, the importance of developing a neutralizing antibody response in the acute phase and its relationship with progression to severe disease or fatal outcome among hospitalized patients remains unclear. In this study, we aim to characterize and compare longitudinally the primary humoral immune host response in the early stages of the disease, looking for an association between neutralization, antibody titers, infective viral lineage, and the clinical outcome in hospitalized and non-hospitalized patients. A total of 111 patients admitted at INER from November 2021 to June 2022 were included. We found that patients with negative or low neutralization showed a significant reduction in survival probability compared to patients with medium or high neutralization. We observed a significant decrease in the median of neutralization in patients infected with viral variants with changes in RBD of the spike protein. Our results suggest that developing an early and robust neutralizing response against SARS-CoV-2 may increase survival probability in critical patients.

## 1. Introduction

The global spread of SARS-CoV-2 infections has resulted in substantial morbidity, mortality, and social disruption worldwide [1]. Patients with moderate and severe COVID-19 develop a wide range of complications, including respiratory failure, acute kidney injury, thrombotic events, cardiovascular damage, and neurologic compromise, which ultimately leads to death [2,3,4]. The host immune response to SARS-CoV-2 appears to play a critical role in disease pathogenesis and clinical manifestations in severe COVID-19 [2,5]. In the initial steps of infection, SARS-CoV-2 activates the immune system and induces a cascade of innate and adaptive immune responses to restrain and control the infection. However, for reasons that are not fully understood, this leads to an uncontrolled and magnified inflammatory response [6].

Adaptive immunity, including humoral and cellular response, plays a critical role in the elimination of pathogens via several mechanisms [5,7,8]. Cytotoxic lymphocytes can eliminate infected cells; B cells form germinal centers can proliferate and differentiate into plasma cells, producing and secreting specific antibodies to control viral replication [9]. Regarding humoral response, specific antibodies against SARS-CoV-2 have the ability to opsonize the pathogen, interact with T cells via antibody-dependent cellular cytotoxicity, activate the complement system cascade, and inhibit or neutralize infection of cells in order to control viral replication, eliminate infected cells and avoid progression of the disease [9].

Among these virus-specific antibodies, neutralizing antibodies (nAb) are those capable of blocking SARS-CoV-2 spike (S) protein interaction with the angiotensin-converting enzyme 2 (ACE2) human receptor blocking viral entry to the host cells [10]. Neutralizing antibodies play an essential role in virus clearance and have been considered key immune products in the protection against or treatment of viral diseases [7,11,12,13,14]. Eliciting a neutralizing-antibody response is a goal of many vaccine development programs and an adequate immunological response is commonly correlated with protection from the disease [8,12].

Several studies showed an association between disease severity and neutralization [15,16]. For instance, Legros et. al showed that nAb titers and anti-spike IgG levels correlated strongly with disease severity, demonstrating that critical patients exhibited high nAb titers, while patients with moderate disease had heterogeneous nAb titers, and asymptomatic or outpatient-care patients had no or low nAbs [17].

On the other hand, the development of a potently neutralizing humoral immunity against SARS-CoV-2 characterized by the presence of neutralizing antibodies within the first weeks from the onset symptoms has been shown to correlate with a decrease in time to a negative swab result and appears to increase survival [18,19]. However, these studies were performed in the sub-acute or later phases. To date, few reports studying the longitudinal dynamics of antibodies in the acute phase of the disease and relating it to the clinical outcome have been published [13,18].

As the pandemic progressed, viral variants sharing a repertoire of single amino acid mutations in the viral genome, including the spike protein, were detected [20,21]. The emergence of viral variants with mutations in the spike protein, specifically in the Receptor Binding Domain (RBD), has altered several properties of the virus, including increased transmissibility, mortality, and the ability to evade the immune system by several mechanisms [22]. Gamma and Beta variants harbor K417N and E484K mutations, Delta harbors T478K and L452R, and Omicron harbors several mutations in RBD, such as K417N, N440K, G446S, G496S, and Q498R [22,23]. 

Between November 2020 and February 2021, Mexico City experienced a dramatic rise in COVID-19 cases and mortality. Derived from genomic surveillance, the emerging viral lineage B.1.1.519 was detected [24]. This lineage possesses three amino acid changes in the spike protein: T478K, P681H, and T732 [24,25]. These mutations can diminish or abolish the neutralizing activity of nAbs or convalescent plasma [22,26]. To date, there is no information regarding neutralization from this lineage.

Until now, the importance of developing a neutralizing antibody response in the acute phase and its relationship with progression to severe disease or fatal outcome among hospitalized patients remains unclear. In this study, we aim to characterize and compare longitudinally the primary humoral immune response of patients infected with distinct lineages of SARS-CoV-2 in the early stages of the disease, looking for an association between neutralization, antibody titers, and the clinical outcome in hospitalized and non-hospitalized patients.

## 2. Materials and Methods

### 2.1. Experimental Design

In order to characterize the neutralizing antibody kinetics in the early stages of the infection, 96 hospitalized patients and 15 non-hospitalized patients were enrolled in the study, with a total of 393 blood samples collected in a period of time between November 2020 and June 2022. All patients had their COVID-19 diagnosis confirmed by RT-qPCR and were treated at the “Instituto Nacional de Enfermedades Respiratorias Ismael Cosío Villegas”. All patients had sequential follow-up blood samples during their hospitalization or ambulatory treatment, with a range of 3–4 longitudinal time points per patient with a time interval of 7–9 days between each measurement. All samples were taken over a period of time from 5–50 days post-symptom onset. 

Demographic data, clinical symptoms, laboratory data, and outcome-related information were obtained by electronic medical records. Clinical management was performed according to the standard of care and clinical criteria for attending physicians. Vaccinated patients, cases of reinfection, and immunosuppressed patients by any cause were excluded from this study.

### 2.2. Neutralization Test and Total Anti-N Antibody Measurement

Neutralization antibodies were detected using a surrogate virus neutralization test (Genscript cPass SARS-CoV-2 Neutralization Antibody Detection, Piscatawa, NJ, USA) following the manufacturer’s recommendations. Briefly, the SARS-CoV-2 Neutralization Antibody Detection Kit is based on a sandwich ELISA methodology. This is a surrogate test designed to mimic the virus–host interaction by direct protein–protein interaction, using a purified receptor-binding domain (RBD) from the S protein and the host cell receptor ACE2. This RBD–ACE2 interaction can be neutralized by specific NAbs in patient sera, in the same manner as in cVNT or pVNT. Briefly, the serum or plasma was incubated with HRP-RBD solution, then the mix was added to the plate pre-coated with hACE2 protein. All HRP-RBD unbind was captured on the plate while the neutralizing antibody HPR-RBD complex remained in the supernatant and was discarded during the washing steps. Afterward, TMB was added, and the reaction turned yellow with the stop solution. The assay was read at 450 nm in an ELISA plate reader. 

We measured total anti-SARS-CoV-2 nucleocapsid (anti-N) antibodies with Elecsys^®^ Anti-SARS-CoV-2 (Roche, Basel, Switzerland), which is an electrochemiluminescence-based assay following the manufacturer’s recommendation on a Cobas e411 instrument.

### 2.3. SARS CoV-2 Variants Characterization

Viral nucleic acid extraction was performed using a MagNa Pure L.C. 2-0 system (Roche, Indianapolis, OH, USA) or QIAamp viral RNA Minikit (Qiagen, Hilden, Germany). Libraries for the whole-genome sequencing of SARS-CoV-2 were generated using a Covid-Seq kit (Illumina, San Diego, CA, USA). Libraries were sequenced on a MiSeq sequencing platform using a 2 × 150-cycle to obtain paired-end reads (Illumina, San Diego, CA, USA). The DRAGEN COVIDSeq Test Pipeline on BaseSpace Sequence Hub performed the analysis, mapping, and consensus.

### 2.4. Neutralization and Total Anti-N Production Categories

To analyze neutralizing antibody results, a neutralization category was made following the Genscript cPass SARS-CoV-2 Neutralization Antibody Detection Kit´s recommendation. The categories are negative (0–30%), low (30–59%), medium (60–90%), and high (>90%) antibody titers. For total anti-N antibodies, we performed categorization according to the manufacturer’s cut-off points: negative (≤0.99), low production (1–10), and high production (≥10).

### 2.5. Statistical Analyses

All statistical analyses were performed by outcome (non-hospitalized, recovered, or deceased), by infective viral lineage (Wuhan, Lineage B.1.1.519, Delta, and Omicron), and by antibody type (total and neutralizing). Since all data followed a non-normal distribution according to the Kolmogorov–Smirnov test, we used the Chi-Square test to compare categorical variables, the Kruskall–Wallis test, and the Wilcoxon rank-sum test for numerical variables.

For neutralizing and total antibodies, we constructed line plots for each group (non-hospitalized, recovered, and deceased) to represent the kinetics of antibody neutralization of each group of patients through the time of the study. Line plots were constructed using the median neutralization value or total anti-N antibodies value through the sequential follow-up measurements.

To characterize longitudinally the neutralizing and anti-N antibody response, line charts with standard deviations were constructed using the median neutralization value and median anti-N antibody value of each study group, and those values were plotted according to every follow-up measurement. Statistically significant differences between groups were addressed with a Kruskal–Wallis test performed in the vegan R package [27].

To analyze the proportions of all antibody titers categories, we utilized the neutralization categories established by the manufacturer and calculated the percentage of patients classified in each category throughout the four follow-up measurements and compared them between severity groups and viral variant groups. To assess if there were statistically significant differences, a Chi-square test was performed.

To detect if antibody titers significantly affected survival probability, Kaplan–Meier curves coupled with a log-rank test were constructed using the survival R package [28] and plotted with the survminer R [29] package. The number of days after the onset of symptoms was used as a time variable for all curves, the outcome (deceased or recovered) as a dependent variable, and the antibody titers category (low, medium, and high for neutralizing antibodies, and low and high for total anti-N) as exposure.

All analyses and graphs were performed in the R program (v.1.3.1) [30].

## 3. Results

### 3.1. Cohort Characteristics

A total of 111 patients that met the selection criteria and were admitted at INER from November 2021 to June 2022 were included in this study. Demographic data and study groups are described in Table 1. Most of the deceased patients were predominantly male (67.5%), while ambulatory patients were predominantly female (66.6%). There were no differences in hospitalization days between deceased and recovered patients (Wilcoxon rank sum test *p* = 0.15).

All non-hospitalized patients were less than 60 years old with a median age of 38, while 48.64% of deceased patients were older than 60 years. Significant differences were also found in comorbidities, where 20% of ambulatory patients had at least one comorbidity. In contrast, 83.7% of deceased patients had at least one comorbidity. (Kruskal–Wallis *p* = 0.041).

Arterial hypertension was the most prevalent comorbidity in deceased patients (45.9%). Diabetes and smoking were also highly prevalent among the cohort (37.8% and 30%), but with no significant differences between groups.

Prognostic and clinical scores were evaluated between groups, and we found a higher median SOFA (Sequential Organ Failure Assessment) score in the deceased patients (*p* = 0.015) with significant differences between groups. 

### 3.2. Severity Dictates Neutralizing Antibody Kinetics 

We analyzed the longitudinal dynamics of neutralizing and total anti-N antibodies of mild and critical COVID-19 patients (Figure 1). Significant differences in the neutralizing antibody analysis were shown in the mild disease group (non-hospitalized) that had a unique pattern, starting with a low inhibition percentage and then increasing through the second and third follow up (15–30 days post-symptom onset), and finally showing a slight decrease in the neutralizing ability after 30 days of symptom onset (Figure 1). Significant differences were observed in the first follow up, representing the first 14 days from symptom onset, and in the last follow up, representing 30 to 60 days from symptom onset (Figure 1). Comparisons and statistical differences are described in Appendix A.

Regarding total anti-N antibodies, statistically significant differences were observed in the second, third and fourth follow-ups (Figure 2). Patients with a mild version of the disease showed higher antibody concentrations than hospitalized patients, increasing progressively through the time of the study. There were no differences between deceased and recovered hospitalized patients (Figure 2).

### 3.3. Neutralization Categories across Severity Groups

We used a categorical neutralization classification according to the cutoff values proposed in the neutralizing antibody detection kit: negative (0–30%), low (30–59%), medium (60–90%), and high (>90%) antibody titers and built a normalized to 100% bar plot to compare the proportion of patients that were classified in each category for each study group. We built a barplot collage of four panels; each panel represents a different longitudinal measurement. (Figure 3) At the initial measurement, corresponding to the first 15 days post-symptom onset, a high proportion of outpatients (86.7%) did not show a positive neutralizing antibody response; 6.7% of outpatients were classified with a medium neutralization category and 6.7% were classified with a high neutralization. Significant differences were observed between deceased and recovered patients (*p* < 0.0001). A higher proportion of deceased patients had a classification of low neutralization (45.9%), compared to recovered patients where only 33.9% presented this classification. Moreover, a proportion of 37.3% of recovered patients was classified with medium neutralization compared to 29.7% of deceased patients; this comparison indicates that recovered patients had a more robust neutralizing antibody response in the first days post-symptom onset compared to deceased patients.

The exact comparisons were made across all follow-up measurements with similar trends, showing that a more significant proportion of recovered patients presented a higher neutralization category across all follow-up measurements (Figure 3).

### 3.4. The lack of a Neutralizing Response Diminishes Survival Probability 

In order to evaluate the impact of neutralization and antibody titers on the survival probability, Kaplan–Meier curves were constructed using the following neutralization categories: negative (0–30%), low (30–59%), medium (60–90%), and high (>90%) antibody titers and total anti-N antibody titers categories (low and high) with the days post-symptom onset as the time variable. Curves were constructed for each follow-up measurement. For the first follow-up, representing the first 14 days post-symptom onset, we found that patients with negative or low neutralization showed a significant reduction in survival probability compared to patients with medium or high neutralization (log rank test, *p* = 0.0002) (Figure 4A).

When analyzed the total anti-N antibody titer categories and we found that a low antibody titer in the first measurement also showed a reduction in survival probability (log rank test, *p* = 0.008) (Figure 4B).

The same analysis was performed for the remaining sequential follow-up measurements (second, third, and fourth) for neutralizing and total antibodies. Regarding neutralization, we found a significant difference in the second follow-up measurement (*p* = 0.02) and we observed a trend in the next follow-up measurements, but it was without statistical significance.

### 3.5. Effect of Mutations in RBD of Spike Protein in Neutralization Kinetics

Since the first case of SARS-CoV-2 in February 2020, Mexico has had four epidemic waves, with the lineage B.1.1.519, VOC Delta (B.1.617.2), and VOC Omicron dominating the second, third, and fourth waves, respectively. According to this, we analyzed the primary neutralizing antibody response in patients infected with distinct viral lineages. We performed a comparison of neutralization kinetics in 96 hospitalized patients divided into four groups according to the viral lineage they were infected with: (1) patients infected with Wuhan-like lineages (N = 36), (2) patients infected with the B.1.1.519 lineage (N = 29), (3) patients infected with the Delta variant (N = 23), and (4) patients infected with the Omicron variant (N = 8). We constructed a line plot showing the median neutralization values and standard error of every follow-up measurement for all viral lineage groups (Figure 5).

When compared the median neutralization values of the four groups and observed a significant difference in all of the four follow-up measurements (Kruskal–Wallis *p* = 0.0069). Differences were characterized by a decrease in neutralization in patients infected with viral variants with numerous amino acid substitutions in the RBD of the spike protein (Delta and Omicron). In sequential follow-up measurements, neutralization values of patients infected with B.1.1.519 were similar to Wuhan-like lineages, but patients infected with Delta and Omicron showed lower median neutralization values across all of the four measurements, with the greatest differences were observed in the second and third follow-up. Comparisons of intergroup and intragroup are shown in the Appendix A. This finding suggests that viral variants with amino acid changes in the RBD exhibit a reduction in neutralization values.

Additionally, we built normalized to 100% bar plots to compare the proportions of patients in each neutralization category: negative (0–30%), low (30–59%), medium (60–90%), and high (>90%) antibody titers (Figure 6). In this plot, we observed that patients infected with Wuhan-WT showed the greatest proportion of patients with neutralizing activity in the first 14 days post-symptom onset. (88.9%) When we compared the proportions with other viral lineages, we found significant differences (4 × 4 Chi-squared test *p* < 0.001). Only 62.5% of patients infected with Omicron exhibited a detectable neutralizing response.

At the second measurement, corresponding to the days 15–21 post-symptom onset, we also detected significant differences among proportions. (4 × 4 Chi-squared test *p* < 0.001). Patients infected with the Wuhan-WT virus resulted in 65% of patients classified with medium neutralization and 35% classified with high neutralizing antibody titers, while patients infected with other lineages showed a distinct pattern. In patients infected with B.1.1. 519, 3.4% had low antibody titers, 51.7% had medium antibody titers, and 37.9% of patients exhibited high neutralizing antibody titers.

Patients infected with Delta resulted in 4.3% of patients with negative production, 26.1% with low antibody titers, 43.5% with negative antibody titers, and 26.1% had high antibody titers. Patients infected with Omicron resulted in 12% of patients with negative antibody titers, 62.5% of patients with low antibody titers, 12.5% with medium antibody titers, and 12.5% with high antibody titers.

In the third and fourth follow-up measurements, we could not find statistically significant differences, but we found a similar trend where patients infected with Delta and Omicron had a greater proportion of patients with low or medium production compared to patients infected with Wuhan or B.1.1.519, where most of the patients achieved a high antibody titers by this point in time (30 or more days post-symptom onset). This finding highlights the relevance of studying the antibody kinetics at the early stages of the disease.

## 4. Discussion

Understanding the immunopathology of severe pulmonary disease and critical illness in patients with COVID-19 is essential to the development of immunotherapies for this condition. Moreover, knowledge of the dynamics of the primary immune response in SARS-CoV-2 infection is still not fully understood. In this study, we analyzed the neutralizing and total antibody response against SARS-CoV-2 infection in a cohort of non-vaccinated patients, comparing the primary humoral response between severity groups and different viral lineages. A correlation between disease severity and antibody titers has been previously described by other authors [16,31,32] where they found higher antibody titers in patients with more severe disease, compared to non-hospitalized patients.

Shreds of evidence point out that hospitalized COVID-19 patients elicit a robust humoral immune response against SARS-CoV-2 between 14 days post-symptom onset. Additionally, nAbs appear earlier and at higher titers in patients with severe or moderate disease than in those with mild disease [16]. 

This evidence is consistent with our findings that clearly indicate a difference in the induction of the neutralizing response between mild and severe-to-critical disease. However, in this study, we observed differences in the antibody titers between deceased and recovered patients. Being the recovered patients, those who developed a more robust antibody response when classifying their neutralization response in the categories described previously in the methodology and results. A greater proportion of recovered patients were classified as “medium” neutralization, while most of the deceased patients had a “low” neutralization classification on the first 14 days post-symptom onset. (Figure 3).

Previous reports have analyzed the antibody response against SARS-CoV-2 in patients ranging from asymptomatic individuals to ICU patients [14,15,33,34]. Roltgen et al made a detailed analysis of antibody responses against RBD, S1, and N antigens. Finding that outpatients with the mildest illness showed higher titers of anti-spike antibodies compared to N suggested that early humoral immune responses focused on spike antigens in order to neutralize the virus, constrain the viral infection, and avoid the progression of the disease [15]. These results were not consistent with ours. Surprisingly, we found greater anti-N antibody titers in mild disease patients when compared with hospitalized patients. Nonetheless, our results are similar to what Shaffer et al. reported [35], when they analyzed the longitudinal course of antibody levels of various isotypes against S protein or N antigen, finding that different patterns of antibody kinetics depended on the target antigen and antibody isotypes. They found that IgA and IgG against the SARS-CoV-2 S-protein showed a decrease at 6 weeks post-symptom onset; however, the total anti-N antibody levels remained stable and increased through the time of study in non-hospitalized patients. Despite the development of antibodies against surface and internal proteins of SARS-CoV-2, viral RNA could still be detected in posterior oropharyngeal saliva samples from a third of patients for 20 days or longer, even in patients with mild COVID-19 [36]. This long-lasting viral load in the oropharynx or bloodstream could be the main factor stimulating B cells and plasma cells to produce an antibody response against the N protein [37].

There are other reports [38] where patients with severe COVID-19 showed a delay in IgM seropositivity when analyzing the antibody response in moderate, severe, and critical patients. In our study, we could not analyze moderate or severe patients, but we found similar results in critical deceased patients. We observed a delay and a decrease in the neutralizing antibody titers on the first days post-symptom onset.

These findings highlight the role of an adequate antibody response in hospitalized patients, especially at the early stages of the disease (first 14 days post-symptom onset) At this point in the disease’s natural history, viral spread and replication can still be contained, avoiding progression to critical disease [19,39].

The finding that critical patients have increased neutralizing antibody titers could be explained by the possibility that uncontrolled viral spread leads to a state of exacerbated inflammation and increased viral antigen load, which will favor humoral response. Roltgen et al. [15] demonstrated that the development of plasma antibodies was correlated with decreases in viral RNAemia, consistent with potential humoral immune clearance of the virus. The degree of neutralization activity in the early stages of the disease may reflect the clearance ability from circulation, limiting viral replication and avoiding progression to critical disease to death. 

The ongoing COVID-19 pandemic has led to the emergence of SARS-CoV-2 variants, which are characterized by their increased transmissibility, disease severity, and ability to evade humoral immunity. To date, currently circulating SARS-CoV-2 variants share several mutations that enable them to spread in the face of rising population immunity while maintaining or increasing their replication fitness. Most of these mutations are in the spike protein gene. Since spike protein mediates the virus attachment to human cell surface angiotensin-converting enzyme 2 (ACE-2) receptor, mutations in this region are considered of high clinical relevance in terms of virulence, transmissibility, and host immune evasion [21,23,40].

In our study, we analyzed the impact of RBD mutations on the primary immune response. We found significant differences in the antibody response according to the viral lineage that was studied. We observed a reduced neutralization in patients infected with viral variants that have several RBD mutations, such as Delta or Omicron. Nevertheless, we also found differences in neutralization in patients infected with B.1.1.519. These findings suggest that specific amino acid changes in the RBD, such as the T478K mutation, may have an impact on the primary immune response. 

A small minority of mutations are expected to impact virus phenotypes in a way that confers a fitness advantage. T478K has been detected in other phylogenetically non-derived lineages from B.1.1.519 and this substitution has been preserved through the SARS-CoV-2 evolutionary divergence, supporting the hypothesis that this mutation may confer a viral fitness advantage of immunological escape [22,23,26].

Our study has several limitations. First, we could not include patients with moderate disease. This study was made in a third-level national reference hospital which, during the pandemic in Mexico, hospitalizations were prioritized for patients with severe to critical disease and ICU requirements. Second is that our experimental methodology could not differentiate between immunoglobulin isotypes. To date, there are several reports, such as Sterling et al., who gave evidence that early SARS-CoV-2-specific humoral responses were dominated by serum IgA [41]. 

As the spectrum of SARS-CoV-2 variants is expanding, laboratory correlates of protection against SARS-CoV-2 variants have become a high priority. Neutralizing antibody titers correlate with protection from infection. Nevertheless, we must not lay aside other aspects of protective immunity, including innate immunity, memory B cells, antibody-mediated effector functions, and T-cell immunity. A greater understanding of the correlates of immune protection is required to provide a strategy to identify those variants that pose the greatest threat to current vaccines and to guide the development of new therapeutic agents, such as antibody-mediated therapies.

## 5. Conclusions

In this cohort of non-vaccinated critical hospitalized patients, the development of an early and robust neutralizing humoral response against SARS-CoV-2 in the first 14 days post-symptom onset appears to increase survival. Our study is the first to describe neutralizing and total antibody response of patients infected with lineage B.1.1.519 and additionally evaluate the impact of different VOCs in the primary immune response to natural infection. Our findings suggest that the viral variant may be one of the factors influencing an early and robust humoral immune response. We hypothesize that there is a critical time window in the first days of infection in which the neutralizing antibody response must be developed adequately in order to control the viral spread and avoid progression to critical disease or fatal outcome.

## Figures and Tables

**Figure 1 vaccines-10-02063-f001:**
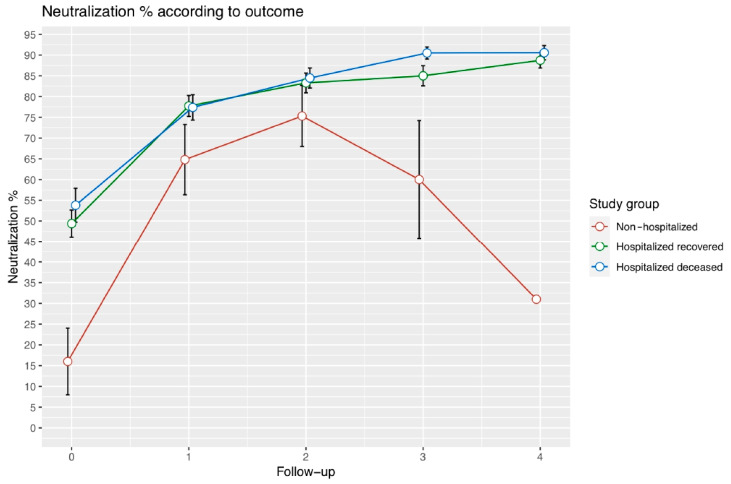
Longitudinal measurements of neutralizing antibodies in different groups of severity. Lineplot representing inhibition percentage kinetics of neutralizing antibodies through four longitudinal measurements. Each point represents the median neutralization value with standard error for that follow-up. The non-hospitalized group is represented by the orange line, the blue line represents recovered hospitalized patients, and the purple line represents deceased hospitalized patients.

**Figure 2 vaccines-10-02063-f002:**
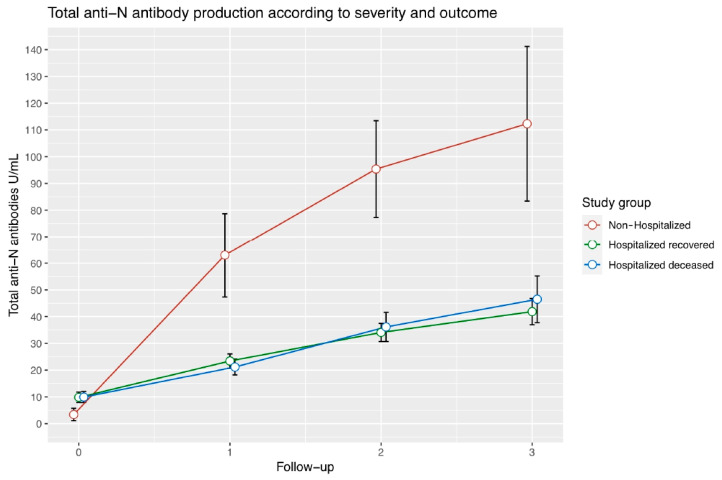
Longitudinal measurements of total anti-N antibodies in different groups of severity. Lineplot representing total anti-N antibody concentration through four longitudinal measurements. Each point represents the median neutralization value with standard error for that follow-up. The non-hospitalized patients group is represented by the orange line, the blue line represents recovered hospitalized patients, and the purple line represents deceased hospitalized patients.

**Figure 3 vaccines-10-02063-f003:**
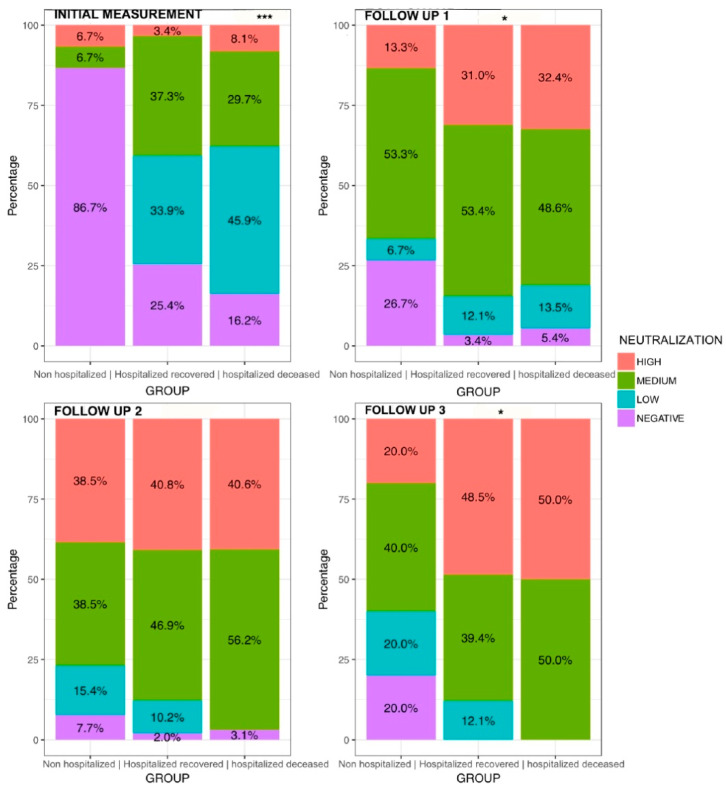
Proportion of patients in each neutralization category. Barplots show the proportion of patients with the different antibody classifications in each follow-up and within study groups (non-hospitalized, recovered, and deceased). Neutralization categories: 0–30% negative production, 31–60% low production, 61–90% medium production, and >90% high production. Initial measurement stands for the first follow-up measurement in a period of <14 days post-symptom onset, follow-up 1 for the period between 14 and 21 days post-symptom onset, follow-up 2 for the period between 22 and 30 days post-symptom onset, and follow-up 3 for the last measurement 30 days post-symptom onset. * = *p* < 0.05, *** = *p* < 0.0001.

**Figure 4 vaccines-10-02063-f004:**
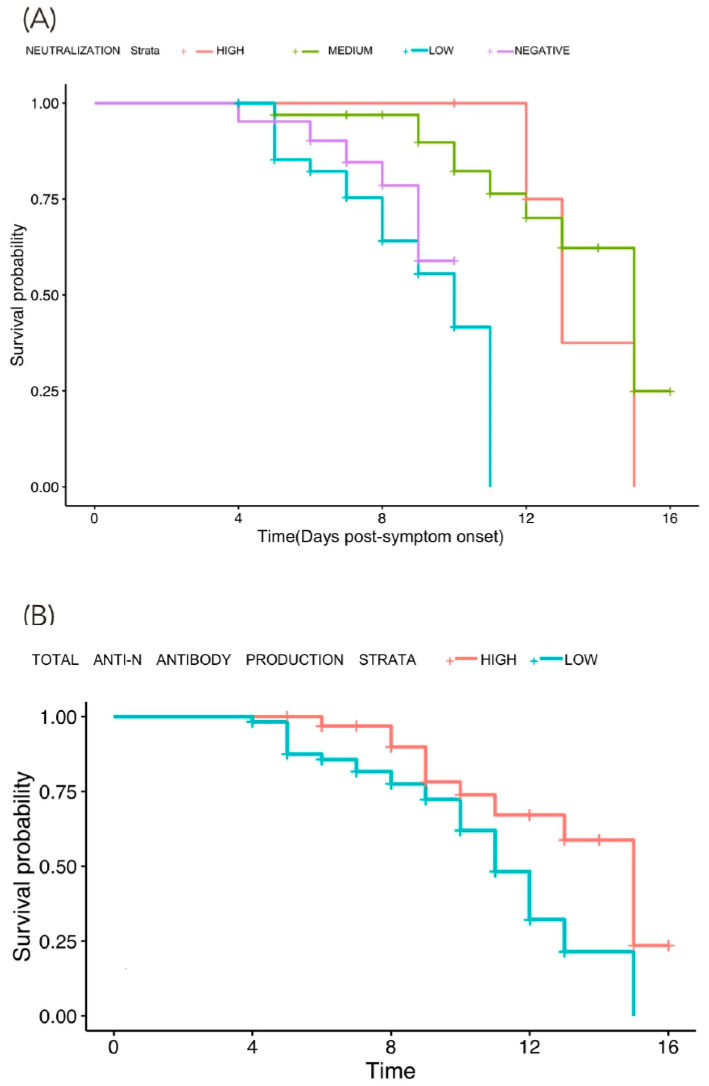
(**A**,**B**). Antibody titers and survival analysis. Kaplan–Meier Curves coupled with the Cox test was constructed to evaluate survival probability according to the neutralization categories (low, medium, and high) and the total anti-N antibody titers (low and high). (**A**): Kaplan–Meier curve showing the neutralization categories in which patients were classified in their initial measurement and the decrease in survival probability in patients with a negative or low antibody production. (**B**): Kaplan–Meier curve showing the total anti-N production classification (high and low) in which patients were classified in their initial measurement and the decrease in survival probability in patients with a low total anti-N antibody production. (**A**) *p* = 0.00025, (**B**) *p* = 0.0359.

**Figure 5 vaccines-10-02063-f005:**
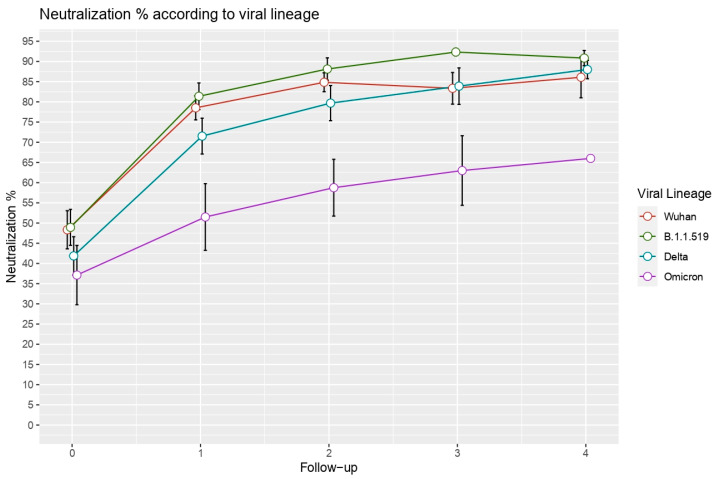
Longitudinal measurement of neutralizing antibodies according to viral lineage. Each point represents the median neutralization value with standard error for that follow-up measurement. Wuhan-like lineages are represented by the orange line, the green line represents patients infected with lineage B.1.1.519, the blue line represents patients infected with VOC Delta, and the purple line represents patients infected with Omicron.

**Figure 6 vaccines-10-02063-f006:**
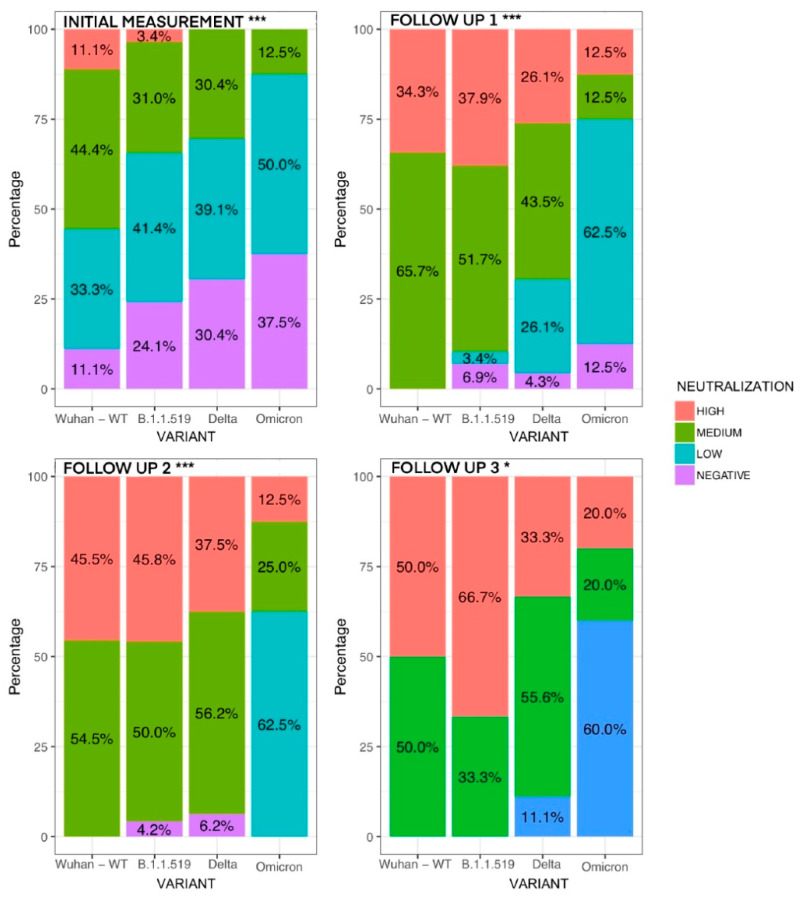
Proportion of patients in each neutralization category barplot collage with four panels, one panel for each of the longitudinal measurements showing the proportion of patients classified according to the following neutralization categories: 0–30% negative production, 31–60% low production, 61–90% medium production, and >90% high production of each viral lineage (Wuhan-WT, B.1.1.519, and VOC Delta). Initial measurement represents a period of <14 days post-symptom onset, follow-up 1 to the period between 14 and 21 days post-symptom onset, follow up-2 to the period between 22 and 30 days post-symptom onset, and follow-up 3 the last measurement (30 days post-symptom onset). * = *p* < 0.05, *** = *p* < 0.001.

**Table 1 vaccines-10-02063-t001:** Demographics of the cohort.

Total Patients = 111	Non-Hospitalized Patients (n = 15)	Hospitalized Recovered Patients (n = 59)	Hospitalized Deceased Patients (n = 37)	Statistical Significance
Men n (%)	5 (33.3%)	35 (59.32%)	25 (67.56%)	*p* < 0.001
Women n (%)	10 (66.6%)	23 (38.98%)	12 (32.43%)	*p* = 0.03813
Edad	38	54	59	*p* < 0.001
Age > 60	0	25 (42.37%)	18 (48.64%)	*p* = 0.0372
Comorbidities	3 (20%)	42 (71.18%)	31 (83.7%)	*p* = 0.04144
BMI	25.5	30.81	28	*p* = 0.12 ^2^
Obesity	4 (26.6%)	31 (52.54%)	17 (45.94%)	*p* = 0.5216 ^1^
DM2	1 (6%)	20 (33.89%)	12 (32.4%)	*p* = 0.2415 ^1^
Arterial hypertension	1 (6%)	18 (30.50%)	17 (45.94%)	*p* = 0.1132 ^1^
Smoking	3 (20%)	18 (30.50%)	14 (37.8%)	*p* = 0.6438 ^1^
Respiratory disease	0	4 (6.77%)	4 (10.81%)	*p*= 0.4281 ^1^
Cardiovascular disease	0	2 (3.38%)	4 (10.81%)	*p* = 0.2182 ^1^
Hospitalization days (median value)	0	23	18	*p* = 0.15 ^3^
*Ventilation*				
IMV	0	45 (76.27%)	37 (100%)	*p* = 0.001648 ^1^
HFNC	0	13 (22.03%)	0	*p* = 0.005278 ^1^
Nasal cannula	0	1 (1.7%)	0	
Without oxygen support	15 (100%)	0	0	
Days of IMV	0	12	18	*p* = 0.18 ^3^
SOFA	0	4	5	*p* = 0.015 ^3^
APACHE II	0	10	12.5	*p* = 0.046 ^3^
*Viral variants*				
Wuhan-WT	15 (100%)	22 (37.28%)	14 (37.83%)	*p* = 0.05354 ^1^
B.1.1.519	0	15 (25.42%)	14 (37.83%)	*p* = 0.06935 ^1^
VOC Delta	0	17 (28.81%)	6 (16.21%)	*p* = 0.0877 ^1^
VOC Omicron	0	5 (8.47%)	3 (8.1%)	*p* = 0.5377 ^1^

^1^ Chi-Square test; ^2^ Kruskal–Wallis test; ^3^ Wilcoxon sum rank sum test.

## Data Availability

The clinical datasets used during the current study are available from the corresponding authors upon reasonable request.

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
