# Peer review of "Longitudinal Characterization of a Neutralizing and Total Antibody Response in Patients with Severe COVID-19 and Fatal Outcomes"

_vaccines, 2022, doi:10.3390/vaccines10122063_

Round 1

Reviewer 1 Report

The abstract is clear however it may be useful if in brackets after "High" and "low" neutralisation that you put a value range as these are subjective.

Line 80: the word "owning" here is confusing, please substitute

Line 85: Please provide a reference for this sentence "Gamma and Beta variants harbor K417N and E484Kmutations, Delta harbor T478K and L452R, finally Omicron harbor several mutations in RBD as K417N, N440K, G446S, G496S, Q498R and describe these mutations for the lay reader.

Line 115: "Vaccinated patients, cases of reinfection, and immunosuppressed patients by any cause 115 were excluded from this study." How did the authors control for possible asymptomatic reinfection over the study time period ?

Methods: Is the cPass kit only validated for Wuhan positive samples? If it is to be used for other VOC then my prediction would be that different standards need to be used. How did the authors adapt the assay to perform equally effectively with sera from individuals infected with different VOCs ?

Line 276: which Omicron ? 

How do the authors rule out infection with more than one VOC over the study period ?

The results section is rather confusing and would benefit from more clarity in how the data graphs are described.

There are simple kits available to distinguish between Ig subtype and the authors should consider undertaking such assays to strengthen the narrative and discussion.

Author Response

Reviewer 1

The abstract is clear however it may be useful if in brackets after "High" and "low" neutralization that you put a value range as these are subjective.

R= Thank you for the observation, we added the value ranges  (line 293-294, line 325, line 378 - 379 )

Line 80: the word "owning" here is confusing, please substitute

R= We change the word owning to” sharing a repertoire of single amino acid”
(line 124) 

Line 85: Please provide a reference for this sentence "Gamma and Beta variants harbor K417N and E484K mutations, Delta harbor T478K and L452R, finally Omicron harbor several mutations in RBD as K417N, N440K, G446S, G496S, Q498R and describe these mutations for the lay reader.

R= We added a reference for this sentence (line 131) and the description of the importance of the mutations is shown in lines 126-129.

Line 115: "Vaccinated patients, cases of reinfection, and immunosuppressed patients by any cause  were excluded from this study." How did the authors control for possible asymptomatic reinfection over the study time period ?

R= Thank you for your observation. For each patient, a direct or indirect interview was made (or with their closest family member) in order to retrieve valuable information about the patient's history. We took special consideration asking for vaccination status, previous infections, and complete medical history. Furthermore, when patients completed their sequential blood sampling, in the initial measurement we could rule out previous asymptomatic infections with the total anti – N antibodies, most of the patients had a negative result in the initial measurement, in cases of reinfection, total anti-N antibodies were positive and were excluded for the study.

Methods: Is the cPass kit only validated for Wuhan positive samples? If it is to be used for other VOC then my prediction would be that different standards need to be used. How did the authors adapt the assay to perform equally effectively with sera from individuals infected with different VOCs ?

R = Thank you for your observation. Indeed, initially the cPass neutralization kit from Genscript used the Wuhan original strain in their methodology, nevertheless, as the pandemic progressed and VOCs appeared, Genscript designed a specific RBD for the different circulating VOCs as an additional component of the cPass Kit. We made an additional analysis comparing the performance of different conjugated RBD (Delta and Omicron) against Wuhan’s original RBD. The results are presented in supplementary material Figures S1 and S2. In this analysis, we compared the median neutralization values in each follow-up, comparing the neutralization value obtained with the Wuhan original RBD and with other RBD conjugated (Delta and Omicron) 

We couldn't find significant differences in neutralization values. Meaning that neutralization results are not significantly affected when analyzing data with Wuhan original RBD sequence. Knowing that the performance of both tests was similar, we decided to use the original Wuhan RBD for all samples. Figures are presented in supplementary material 

Line 276: which Omicron ? 

R = The patients were recruited in January 2022 and the sequences confirm they were infected with lineage Omicron BA.1 and BA.1.1

How do the authors rule out infection with more than one VOC over the study period ?

R= When patients are admitted to the Hospitalization area, a diagnostic nasopharyngeal swab is taken and processed for diagnostic procedures additionally as part of the national genomic surveillance program this same sample is taken to the laboratory in order to obtain a whole genome sequence and report the results to GISAID. Furthermore, during patients' hospital stay, sequential nasopharyngeal swabs are taken and are also sequenced and reported to GISAID. This way we can rule out infection with more than one VOC during their hospitalization. 

The results section is rather confusing and would benefit from more clarity in how the data graphs are described.

R= We checked the results section in order to improve the clarity of the data graph description.

There are simple kits available to distinguish between Ig subtypes and the authors should consider undertaking such assays to strengthen the narrative and discussion.

R = We appreciate your observation, and we have previously considered including this type of assay to distinguish between Ig subtypes, nevertheless one of the aims of this project was to evaluate the whole neutralizing response of the individual, taking into consideration all neutralizing immunoglobulins. It is possible that in further studies we can make a detailed characterization of the Ig subtypes that are deeply involved in the neutralizing response.

Reviewer 2 Report

The manuscript by Serna-Muñoz et al. compares and contrasts clinical and demographic features of CoV-2 patients who (a) did not require hospitalization, (b) were hospitalized but non-deceased, or (c) were hospitalized and deceased.   In particular, the authors focus on differences in neutralizing antibody responses between the groups and clinical outcomes.  The underlying precept centers around the recognition (L77-79) that, "few reports studying the longitudinal dynamics of antibodies in the acute phase of the disease relating it with the clinical outcome have been published [13,18]".  The authors conclude (L35-36) that, "developing an early and robust neutralizing response against SARS-CoV-2 may increase survival probability in critical patients".

The fundamental goal of the manuscript is sound and appropriate.  However, there are several flaws that impede critical analysis and acceptance of the fidelity of the results.  In particular, the authors refer to results in the text that are different from the numbers in Table 1.  

The authors state (L181-182) that, "Most of the deceased patients were predominantly male (64.7%)".  However, the data in Table 1 indicates that 67.6% of the deceased patients were male.

The authors state (L190) that, "50% of deceased patients were older than 60 years".  However, the data in Table 1 indicates that 48.6% of the deceased patients were >60 years.

The authors state (L191-192) that, "where 20% of ambulatory patients had at least one comorbidity, in contrast, 85% of deceased patients had at least one comorbidity".  However, the data in Table 1 do not support the assertion that 85% of patients in hospitalized patients had >1 comorbidity.

The authors state (L194) that, "Arterial hypertension was the most prevalent comorbidity in these groups (47%)".  It is not clear to the reader which group the statement refers to since the data in Table 1 for arterial hypertension (6, 30.5, and 45.9%) do not support the authors' statement.

The authors state (L195) that, "Diabetes and smoking were also highly prevalent among the cohort (35% and 29%)".  It is not clear whatsoever how the data in Table 1 support this statement.

There also appears to be a fundamental flaw in the approach to this study.  Notably, the study appears to be base on an unvaccinated study group.  It is not evident in the text whether the cohort were indeed unvaccinated.  If they are unvaccinated, then it is not surprising that there are differences in the neutralization results based on infection with successive CoV-2 variants and the singular test reagent to quantify neutralization.  Given that CoV-2 variants exhibit varying differences in the RBD domain, the assays used by the authors reflects only the binding of patient sera to the prototype[ical RBD sequence used for all samples.  The assay measures binding and does not reflect whether the patient sera bound to the RBD sequence of the particular variant that infected the patient.  Put another way, the assay does not appear to reflect whether they developed neutralizing antibodies to the variant with which they were infected.  Instead, the assay only reflects binding to an RBD sequence that may have no relevance with their clinical outcomes.  The results might conceivably be relevant if the study group had been vaccinated against the prototypical Wuhan strain.

Author Response

Reviewer 2

The manuscript by Serna-Muñoz et al. compares and contrasts clinical and demographic features of CoV-2 patients who (a) did not require hospitalization, (b) were hospitalized but non-deceased, or (c) were hospitalized and deceased.   In particular, the authors focus on differences in neutralizing antibody responses between the groups and clinical outcomes.  The underlying precept centers around the recognition (L77-79) that, "few reports studying the longitudinal dynamics of antibodies in the acute phase of the disease relating it with the clinical outcome have been published [13,18]".  The authors conclude (L35-36) that, "developing an early and robust neutralizing response against SARS-CoV-2 may increase survival probability in critical patients".

The fundamental goal of the manuscript is sound and appropriate.  However, there are several flaws that impede critical analysis and acceptance of the fidelity of the results.  In particular, the authors refer to results in the text that are different from the numbers in Table 1.  

The authors state (L181-182) that, "Most of the deceased patients were predominantly male (64.7%)".  However, the data in Table 1 indicates that 67.6% of the deceased patients were male.The authors state (L190) that, "50% of deceased patients were older than 60 years".  However, the data in Table 1 indicates that 48.6% of the deceased patients were >60 years.The authors state (L191-192) that, "where 20% of ambulatory patients had at least one comorbidity, in contrast, 85% of deceased patients had at least one comorbidity".  However, the data in Table 1 do not support the assertion that 85% of patients in hospitalized patients had >1 comorbidity.The authors state (L194) that, "Arterial hypertension was the most prevalent comorbidity in these groups (47%)".  It is not clear to the reader which group the statement refers to since the data in Table 1 for arterial hypertension (6, 30.5, and 45.9%) do not support the authors' statement.The authors state (L195) that, "Diabetes and smoking were also highly prevalent among the cohort (35% and 29%)".  It is not clear whatsoever how the data in Table 1 support this statement.

 R = Thank you for your observation. We made a mistake, at the final revisions we found out that one patient had previously developed an anti-N antibody response, meaning this patient had a previous asymptomatic infection, thus this patient was eliminated from the final analysis, and we did not update the data described in table 1. 

We made changes in order to ensure our results in the text are consistent with what table 1 shows. (lines 192, 194, 203, 205, 207, 208) 

There also appears to be a fundamental flaw in the approach to this study.  Notably, the study appears to be based on an unvaccinated study group.  It is not evident in the text whether the cohort were indeed unvaccinated.  If they are unvaccinated, then it is not surprising that there are differences in the neutralization results based on infection with successive CoV-2 variants and the singular test reagent to quantify neutralization.  

R= One of the exclusion criteria of this cohort was being partially or fully vaccinated or having a previous SARS-CoV-2 infection. This is because one of the main aims of this study was to evaluate the primary immune response to infection. 

In this study, patients that recognized a previous SARS-CoV-2  infection were eliminated from this cohort. In cases of asymptomatic infection or unconfirmed diagnosis, we eliminated patients that had developed a detectable anti-N antibody response in their initial measurement (first 14 days post symptom onset)

Given that CoV-2 variants exhibit varying differences in the RBD domain, the assays used by the authors reflect only the binding of patient sera to the prototypical RBD sequence used for all samples.  The assay measures binding and does not reflect whether the patient sera bound to the RBD sequence of the particular variant that infected the patient.  Put another way, the assay does not appear to reflect whether they developed neutralizing antibodies to the variant with which they were infected.  Instead, the assay only reflects binding to an RBD sequence that may have no relevance with their clinical outcomes.  The results might conceivably be relevant if the study group had been vaccinated against the prototypical Wuhan strain.

R= Thank you for your observation. B cells response against antigen such as RBD, produce antibodies against multiple epitopes from different regions of the protein. So, a SARS-CoV-2 RBD, contains many different epitopes against which neutralizing antibodies can be elicited (polyclonal response). 

However, some mutations in RBD harboring some variants specially omicron, could reduce the neutralization but not totally abolish these antibody functions. This is supported by experimental data, in supplementary material Figures S1 and S2. We performed neutralization assays with Delta and Omicron RBDs and made a comparison of neutralization values. In this additional assay, we compared the median neutralization values of each follow-up measurement with a coupled Wilcoxon rank sum test and we couldn't find statistically significant differences between RBDs. This finding means that although amino acid changes in the RBD can decrease antibody binding to those specific epitopes, many other antibodies targeting different epitopes will maintain their neutralizing capacity blocking the interaction between RBD and hACE2.  Regarding these results, we decided to continue with the experimental process using Wuhan´s original RBD.

Reviewer 3 Report

Dear Authors,

my congratulations to this comprehensive, clear and well written manuscript based on a well-designed study!

My only comment - and in fact for me a requirement to recommend acceptance of your manuscript for publication - is that you in more detail explain the differences of "neutralizing antibodies" when tested by a neutralization test or by a surrogate neutralization test and reference publications on the comparability of those two analysis methods.

Please clearly state that you did not determine neutralizing activity of SARS-CoV-2 antibodies, but used a sandwich-test which is a surrogate neutralization test, although the test you used is called a "Neutralization Antibody detection Kit". 

Best wishes!

Author Response

Reviewer 3

Dear Authors,

my congratulations to this comprehensive, clear and well written manuscript based on a well-designed study!

My only comment - and in fact for me a requirement to recommend acceptance of your manuscript for publication - is that you in more detail explain the differences of "neutralizing antibodies" when tested by a neutralization test or by a surrogate neutralization test and reference publications on the comparability of those two analysis methods.Please clearly state that you did not determine neutralizing activity of SARS-CoV-2 antibodies, but used a sandwich-test which is a surrogate neutralization test, although the test you used is called a "Neutralization Antibody detection Kit". Best wishes!

R = Thank you for your comment, we will include a brief explanation in the methodology section explaining the differences between SARS-CoV-2 Surrogate Virus Neutralization Test (sVNT) Kit and direct neutralizing antibody detection. (lines 124-128) 

Round 2

Reviewer 1 Report

Fig 4 legend is incomplete.

RE Refs 28-30

28. Therneu, P.M. A Package for Survival Analysis in R 2020. 548

29. Kassambara, A.; Kosinski, M.; Biecek, P. Survminer: Drawing Survival Curves Using ggplot2 2021. 549

30. Team, R.C. R: A Language and Environment for Statistical Computing 2020.

Are these books ? If so they need publisher, chapter info, pages and ISBN. Please provide corrected references here. Also provide weblinks as appropriate

Author Response

Reviewer 1

Fig 4 legend is incomplete.

Response: Thank you for your observation, we completed the legend in figure 4 (lines 280-287). 

RE Refs 28-30

  1. Therneu, P.M. A Package for Survival Analysis in R 2020. 548
  2. Kassambara, A.; Kosinski, M.; Biecek, P. Survminer: Drawing Survival Curves Using ggplot2 2021. 549
  3. Team, R.C. R: A Language and Environment for Statistical Computing 2020.

Are these books ? If so they need publisher, chapter info, pages and ISBN. Please provide corrected references here. Also provide weblinks as appropriate

Response: Thank you for your observation. References 28 and 29 correspond to R packages, and the R program itself (30). Citations 28 and 29 are manuals describing the package, the commands and functions that are included in that package. Citation 30 corresponds to the citation of the R program which is a free software environment for statistical computing and graphics. We considered the R CRAN instructions for citation of R program and R packages for this purpose. We added appropiate weblinks to these references (lines 555-560).